



# Measurement report: Emission factors of NH₃ and NHₓ for wildfires and agricultural fires in the United States

Laura Tomsche[1,2,3,+], Felix Piel[4,5,6], Tomas Mikoviny[5], Claus J. Nielsen[5], Hongyu Guo[7], Pedro Campuzano-Jost[7], Benjamin A. Nault[8], Melinda K. Schueneman[7], Jose L. Jimenez[7], Hannah Halliday[9], Glenn Diskin[2], Joshua P. DiGangi[2], John B. Nowak[2], Elizabeth B. Wiggins[1,2], Emily Gargulinski[10], Amber J. Soja[2,10], and Armin Wisthaler[4,6]

[1]Universities Space Research Association, Columbia, MD, USA
[2]NASA Langley Research Center, Hampton, VA, USA
[3]Institute of Atmospheric Physics, German Aerospace Center, Oberpfaffenhofen, Germany
[4]Department of Chemistry, University of Oslo, Oslo, Norway
[5]IONICON Analytik GmbH, Innsbruck, Austria
[6]Institut für Ionenphysik und Angewandte Physik, Universität Innsbruck, Innsbruck, Austria
[7]Department of Chemistry and Cooperative Institute for Research in Environmental Sciences (CIRES), University of Colorado, Boulder, CO, USA
[8]Center for Aerosol and Cloud Chemistry, Aerodyne Research, Inc., Billerica, MA, USA
[9]US Environmental Protection Agency, Durham, NC, USA
[10]National Institute of Aerospace, Hampton, VA, USA
[+]now at: Institute of Atmospheric Physics, Johannes-Gutenberg University Mainz, Mainz, Germany

*Correspondence to*: Armin Wisthaler (armin.wisthaler@kjemi.uio.no)

**Abstract.** During the 2019 Fire Influence on Regional to Global Environments and Air Quality (FIREX-AQ) study, the NASA DC-8 carried out *in situ* chemical measurements in smoke plumes emitted from wildfires and agricultural fires in the contiguous US. The DC-8 payload included a modified proton-transfer-reaction time-of-flight mass spectrometer (PTR-ToF-MS) for the fast measurement of gaseous ammonia (NH₃) and a high-resolution time-of-flight aerosol mass spectrometer (AMS) for the fast measurement of submicron particulate ammonium (NH₄⁺). We herein report data collected in smoke plumes emitted from six wildfires in the Western US, two prescribed grassland fires in the Central US, one prescribed forest fire in the Southern US, and 66 small agricultural fires in the Southeastern US. Smoke plumes contained double to triple digit ppb levels of NH₃. In the wildfire plumes, a significant fraction of NH₃ had already been converted to NH₄⁺ at the time of sampling (≥2 h after emission). Substantial amounts of NH₄⁺ were also detected in freshly emitted smoke from corn and rice field fires. We herein present a comprehensive set of emission factors of NH₃ and NHₓ, with NHₓ = NH₃ + NH₄⁺. Average NH₃ and NHₓ emission factors for wildfires in the Western US were 1.86 ± 0.75 g kg⁻¹ of fuel burned and 2.47 ± 0.80 g kg⁻¹, respectively. Average NH₃ and NHₓ emission factors for agricultural fires in the Southeastern US were 0.89 ± 0.58 g kg⁻¹ and 1.74 ± 0.92 g kg⁻¹, respectively. Our data show no clear inverse correlation between modified combustion efficiency (MCE) and NH₃ emissions. Importantly, we found that NH₃ emissions in ambient sampling were significantly higher than observed in previous laboratory experiments with similar fuel types.



## 1 Introduction

Ammonia ($NH_3$) is an important trace gas in the Earth's atmosphere that is mostly emitted from agriculture, traffic, the oceans and biomass burning. In the presence of acids, $NH_3$ rapidly partitions to aerosol particles, which in turn impact air quality and climate (Seinfeld and Pandis, 2016). In much of the atmosphere, $NH_3$ exhibits a major influence on particle acidity (pH), which

is a major controlling parameter for many important aerosol physical and chemical processes (*e.g.*, Pye et al., 2020; Nault et al., 2021). $NH_3$ is also the largest contributor to deposition of nitrogen from the atmosphere to soil and vegetation, causing surface water eutrophication, soil acidification, and ultimately biodiversity loss (*e.g.*, Bobbink and Higgs, 2014).

Fires emit $NH_3$ predominantly during smoldering combustion, which occurs at low temperatures (*e.g.*, Lobert et al., 1990; Yokelson et al., 1996, 1997; Goode et al., 1999; McMeeking et al., 2009; Burling et al., 2010; Roberts et al., 2020). $NH_3$ is

typically the third most abundant nitrogen compound (after $N_2$ and NO) and the most abundant reduced nitrogen compound emitted from fires (Lobert et al., 1990; Roberts et al., 2020; Lindaas et al., 2021).

An important parameter for investigating the atmospheric impact of $NH_3$ is the emission factor, $EF_{NH_3}$, which is the mass of $NH_3$ (in g) that is emitted per mass of fuel burned (in kg). Several literature reviews (Andreae and Merlet, 2001; Akagi et al, 2011; Andreae, 2019; Prichard et al., 2020) report $EF_{NH_3}$ values for different types of fire fuels. A closer look at the literature

reveals that emissions from fuels that are typical of the United States (US) have mostly been studied in the laboratory (*e.g.,* Yokelson et al., 1996; McMeeking et al., 2009; Burling et al., 2010; Stockwell et al., 2015; Selimovic et al., 2018; Roberts et al., 2020). Previous work has shown that laboratory fires may not realistically simulate fires occurring in the real world due to different burning conditions and the lack of heterogeneity in fuels (*e.g.*, Yokelson et al., 2013; Hodshire et al., 2019). Only very few studies have reported $EF_{NH_3}$ derived from measurements carried out in the field (*e.g.*, Lindaas et al., 2021).

The limited availability of field data is mostly because $NH_3$ is difficult to measure. $NH_3$ is a "sticky" molecule that easily adsorbs onto inlet and instrumental surfaces. This makes fast airborne measurements of $NH_3$ particularly challenging. Müller et al. (2014; see Supplement) have shown that proton-transfer-reaction time-of-flight mass spectrometry (PTR-ToF-MS) can be used for airborne $NH_3$ measurements, although with some limitations tied to a relatively slow time response and a poor detection limit due to a large intrinsic background. The University of Innsbruck PTR-ToF-MS instrument has been used for

airborne measurements of $NH_3$ in previous studies (Sun et al., 2015; Kelly et al., 2018; Guo et al., 2021; Da Pan et al., 2021).

The Fire Influence on Regional to Global Environments and Air Quality (FIREX-AQ) study was a joint NOAA/NASA effort to investigate the atmospheric impact of wildfires and agricultural fires in the contiguous US (Warneke et al., 2022). In summer 2019, the NASA DC-8 Airborne Science Laboratory performed *in situ* measurements in smoke plumes emitted from wildfires in the Western US and agricultural fires in the Southeastern US. The aircraft payload included a PTR-ToF-MS instrument that

was modified and optimized for the fast measurement of $NH_3$. It also included an aerosol mass spectrometer (AMS) for fast measurement of submicron particulate ammonium ($NH_4^+$). This allowed us to measure and report a set of emission factors of $NH_3$ and $NH_x$, with $NH_x = NH_3 + NH_4^+$, for different types of fires.





## 2 Methods

### 2.1 FIREX-AQ

The FIREX-AQ experiment has been described in detail by Warneke et al. (2022). During the 2019 field campaign, $NH_3$ and $NH_4^+$ were measured aboard the NASA DC-8 in smoke plumes emitted from six wildfires in the Western US (Shady, Williams Flats, Castle, Ridge Top, Mica/Lick Creek, Horsefly), two prescribed grassland fires in the Central US (Hickory Ridge State Wildlife Management Area, Tallgrass Prairie National Preserve), and one prescribed forest fire in the Southern US (Black Water River State Forest). A map showing the location of these fires is given in the Supplement (Fig. S1). Vegetation and fuel

type information is summarized in Tab. S1. Several downwind transects were typically flown in the smoke plumes emitted from the wildfires. In addition, the NASA DC-8 sampled smoke plumes from a large number of agricultural fires in the Southeastern US. These small plumes were typically sampled twice in perpendicular direction. We successfully measured $NH_3$ and $NH_4^+$ in plumes emitted from 66 agricultural fires.

### 2.2 Instrumentation

A modified PTR-ToF-MS instrument was used for fast-response measurements of $NH_3$ aboard the NASA DC-8 during FIREX-AQ. The airborne PTR-ToF-MS analyzer has been described in detail by Müller et al. (2014). Only the modifications pertinent to the fast measurement of $NH_3$ are thus described here.

For reducing the instrumental $NH_3$ background, 12-25 standard cubic centimeters per minute (sccm; "standard" herein means referenced to a temperature of 273.15 K and a pressure of 101325 Pa) of ultra-pure helium (6.0; Praxair Inc., Danbury, CT,

US) were introduced into the source drift region between the drift tube and the ion source. This reduced the backflow of nitrogen into the plasma region and suppressed $NH_3$ formation in the plasma (Müller et al., 2020, and references therein). The instrumental $NH_3$ background was thereby reduced from triple digit to low single digit ppb levels.

For improving the instrumental time response to $NH_3$, all stainless-steel parts in the drift tube were surface-passivated with a functionalized hydrogenated amorphous silicon coating (Piel et al., 2021), and the drift tube was heated to 120 °C. Surface

passivation and heating significantly reduces the adsorption of $NH_3$ to instrumental surfaces, lowering the instrumental response time to ~2 seconds (see Figure 4 of Piel et al., 2021).

A series of inlet configurations were tested during the initial phase of the FIREX-AQ campaign. The fastest response to $NH_3$ was achieved when air was sampled at a flow rate of ~60 standard liters per minute (slpm) through a heated Teflon PFA tube (length: ~2 m, inner diameter: 3.96 mm, wall temperature: 60 °C). Evaporation of ammonium nitrate particles in the main

sampling line was not investigated, but is believed to be small due to the short sample residence time (<25 ms). For inertially separating particles from the analyte air, a small flow was subsampled from the main inlet line in rearward direction and directed into the drift tube through a Teflon PFA tube (length: ~10 cm, outer diameter: 3.175 mm, temperature: 120 °C). The subsampling flow was set to ~250 sccm via a pinch valve applied on the PFA tube. An $NH_3$ time response of a few seconds was ultimately achieved (see Results section).



Since in-field calibrations with different methods (NH$_3$ in N$_2$ standard gas cylinder, permeation calibration device, cross-calibration with a Picarro G2103 NH$_3$ analyzer) were inconsistent, we carried out an extensive post-mission NH$_3$ calibration in the laboratory. For that purpose, an artificial atmosphere (NH$_3$ in air) was generated in a 250 L environmental ("smog") chamber equipped with a Fourier Transform Infrared (FT-IR) spectrometer (Bruker IFS 66v/S). The concentrations of NH$_3$ (accuracy: ±5 %) were determined from the FT-IR spectra (120 m path length, 0.125 cm$^{-1}$ spectral resolution) in a global non-

linear least squares spectral fitting procedure (Griffith, 1996) employing the absolute cross sections of NH$_3$ (Gordon et al., 2017). The estimated accuracy of the reported NH$_3$ mixing ratios is ±15 %. We note that this accuracy estimate is not valid when NH$_3$ mixing ratios abruptly changed and inlet/instrument surfaces were not equilibrated.

Submicrometer (50% cutoff size for a vacuum aerodynamic diameter ~1 µm (about 850 nm geometric diameter for most fire plumes based on in-field calibrations) NH$_4^+$ was measured by an Aerodyne high-resolution time-of-flight AMS instrument

(DeCarlo et al., 2006; Canagaratna et al., 2007), with a time resolution of up to 10 Hz time. The accuracy (2σ) of the NH$_4^+$ data is estimated to be ±34 % (Bahreini et al., 2009), while the detection limit was typically much smaller (25 ppt at 1 Hz in clean air, ~200 ppt in fire plumes). The inlet flow was optimized to allow for near real time sampling (0.3 s residence time) and to minimize particle volatilization in the inlet. We note that, based on the current state of knowledge, the AMS NH$_4^+$ data collected in fresh smoke plumes suffer from a minor (≤ 20%) positive interference from reduced organic nitrogen compounds.

A general correction is still under development based on positive matrix factorization (PMF) analysis.

Carbon monoxide (CO) and methane (CH$_4$) were measured by the Differential Absorption Carbon Monoxide Measurement (DACOM) instrument (Sachse et al., 1991), which is based on mid-infrared wavelength modulation spectroscopy. The uncertainty in the CO data is 2.1 ± 0.2 ppb; the uncertainty for CH$_4$ is about 1 %. Carbon dioxide (CO$_2$) was measured by a LICOR model 7000 analyzer (Vay et al., 2009), which is based on nondispersive infrared absorption spectroscopy. For CO$_2$ <

500 ppm, the accuracy is 0.25 ppm and the precision is 0.1 ppm, while for higher mixing ratios the total uncertainty is about 2 %. Information about fuel types was obtained from the 30 m Fuel Characteristic Classification System (FCCS; Ottmar et al., 2007), the 30 m Cropland Data Layer classification 2019 dataset, and ground intelligence.

**2.3 Emission factor, modified combustion efficiency**

We used the carbon mass balance method for calculating $EF_{NH_3}$ (Yokelson et al., 1996; 1999). The underlying assumption is

that the carbon in the fire fuel is predominantly emitted as CO$_2$, CO and CH$_4$. $EF_{NH_3}$ (in g kg$^{-1}$) is thus described by the simplified equation (1):

$$EF_{NH_3} = \frac{\Delta NH_3}{\Delta CO_2 + \Delta CH_4 + \Delta CO} \times \frac{17}{12} \times F_c \times 1000 \quad (1)$$

Δ is the above background mixing ratio in the plume of the respective trace gas, 17 is the molar mass of NH$_3$ (in g mol$^{-1}$), and 12 is the molar mass of carbon (in g mol$^{-1}$). $F_c$ is the fraction of carbon in the fuel, which we assumed to be 0.5 (Yokelson et

al., 1999).



A problem in the calculation of $EF_{NH_3}$ arises from the fact that $NH_3$ is a "sticky" compound. When the aircraft first penetrates a smoke plume, $NH_3$ molecules typically adsorb onto inlet and instrumental surfaces, thereby delaying the signal response of the analyzer. When the airplane exits the plume, the desorbing $NH_3$ molecules cause a signal tailing (Figure S2a). For calculating $\Delta$, we thus applied the method described in the Supplement of Müller et al. (2016) and calculated cumulative

volume mixing ratios including the period after the plume encounter when the $NH_3$ signal tailed off (Figure S2b). The signal tailing was particularly pronounced during the initial phase of the campaign (before 24 August 2019) when the inlet configuration had not yet been optimized. $NH_x$ is the sum of $NH_3$ and $NH_4^+$; $EF_{NH_x}$ was calculated as the sum of $EF_{NH_3}$ and $EF_{NH_4^+}$. The modified combustion efficiency (MCE) was calculated as $\Delta CO_2/(\Delta CO_2+\Delta CO)$.

Data from 180 plume transects were included in our analysis of the wildfire emissions. We only used data from plume transects

in which CO mixing ratios exceeded 300 ppb for more than 20 seconds and from plumes in which MCE values were stable (standard deviation of MCE < 0.05). Data from seven plume transects were excluded due to missing $NH_3$, $NH_4^+$ or $CH_4$ data. Our EF analysis was not based on a single plume transect in closest proximity to the wildfire, as we observed in several plumes that $\Delta NH_3/\Delta CO$ increased during a few initial downwind transects (see Fig. S3). The reason for this increase (typically <15%) is unclear. We thus included all plume transects in our analysis, up to where $\Delta NH_3/\Delta CO$ reached its maximum and derived an

average $EF_{NH_3}$ and $EF_{NH_x}$ ($\pm$ standard deviation, SD). Data from 164 plume transects were included in our analysis of the agricultural fire emissions. Data from 12 plume transects were excluded due to missing $NH_3$ or $NH_4^+$ data.

## 3 Results and Discussion

### 3.1 Airborne measurements of $NH_3$ in smoke plumes

Figure 1a shows the mixing ratio of $NH_3$ (in red) as measured by the PTR-ToF-MS instrument on 7 August 2019 aboard the

NASA DC-8. The aircraft flew nine downwind transects at an altitude of 5160 m above sea level (ASL) for sampling the plume emitted from the Williams Flats Fire in Washington State. The $NH_3$ signal increased with CO (in black) when the plane entered the plume, exhibited a similar time trend as CO within the plume and decreased to background levels outside the plume, although with some tailing (few minutes). $NH_3$ maxima ranged from 110 to 200 ppb, which were typical maximum $NH_3$ levels measured in fire plumes throughout the 2019 FIREX-AQ field campaign. Also shown in Figure 1a is the time trace of $NH_4^+$

(in dark yellow) as measured by the AMS instrument, with maximum mixing ratios ranging from 42 to 65 ppb. The observation of significant amounts of $NH_4^+$ indicates that $NH_4^+$ was primary emitted (as for example observed by Lewis et al., 2009) and/or gaseous $NH_3$ had been partly converted to particulate $NH_4^+$ by the time of sampling ($\geq$2 h after emission). A rapid conversion can be caused by the fast reaction of $NH_3$ with primary emitted acids such as hydrochloric acid (HCl), nitric acid ($HNO_3$) and organic acids, or occur more slowly downwind via the reaction of $NH_3$ with secondary formed acids.



Figure 1b shows the mixing ratios of $NH_3$, $NH_4^+$ and CO as observed when the NASA DC-8 crossed a plume emitted from a small cornfield fire in the Mississippi River Valley on 26 August 2019 at an altitude of 325 m ASL. All data are shown at the frequency they were recorded (5 Hz), which resulted in an increased noise for $NH_3$. The tailing was however reduced to a few seconds with the improved PTR-ToF-MS inlet. We show the 5-Hz data for demonstrating that we succeeded in measuring such small fire plumes from a jet aircraft. For further analysis, we used the 1 second integrated data. Notably, the AMS

instrument detected significant amounts of $NH_4^+$ in this very fresh plume, indicating that either direct emission from the fire or a rapid conversion of $NH_3$ to $NH_4^+$ had occurred. The latter could be caused by the fast reaction of $NH_3$ with HCl, which is emitted in significant amounts from agricultural fires (Liu et al., 2017). Another plausible explanation is the resuspension of recently applied ammonium nitrate fertilizer.

**Figure 1: Mixing ratios of $NH_3$, $NH_4^+$, and CO as measured when the NASA DC-8 transected (a) the plume emitted from the Williams Flats Fire on 7 August, 2019 in downwind direction and (b) the plume emitted from a small corn field fire in the Mississippi River Valley on 26 August, 2019.**




Due to the fact that $NH_4^+$ was already present in very fresh smoke (due to direct emission or rapid conversion), we will herein

also report $EF_{NH_x}$, as suggested in previous work by Hegg et al. (1990). In Figure 2a, we plot $EF_{NH_x}$ against $EF_{NH_3}$ for the six

Western US wildfires investigated during the 2019 FIREX-AQ field campaign. The two EFs are highly correlated ($R^2 = 0.96$),

with the slope of the linear regression curve being close to unity ($1.07 \pm 0.05$). This regression analysis suggests that $NH_4^+$

added ~0.5 g kg$^{-1}$ (offset of the regression line: $0.47 \pm 0.11$) to $EF_{NH_x}$ throughout the campaign. The offset may be interpreted

as the typical direct $NH_4^+$ emission factor (or fast conversion of $NH_3$).

In the case of the agricultural burns, the NASA DC-8 sampled the plumes in very close proximity to the fires. $EF_{NH_x}$ and

$EF_{NH_3}$ had again a regression slope of ~1. The offset was mainly caused by elevated $NH_4^+$ emissions and low $NH_3$ emissions

from some of the cornfield fires (Fig. 2b).

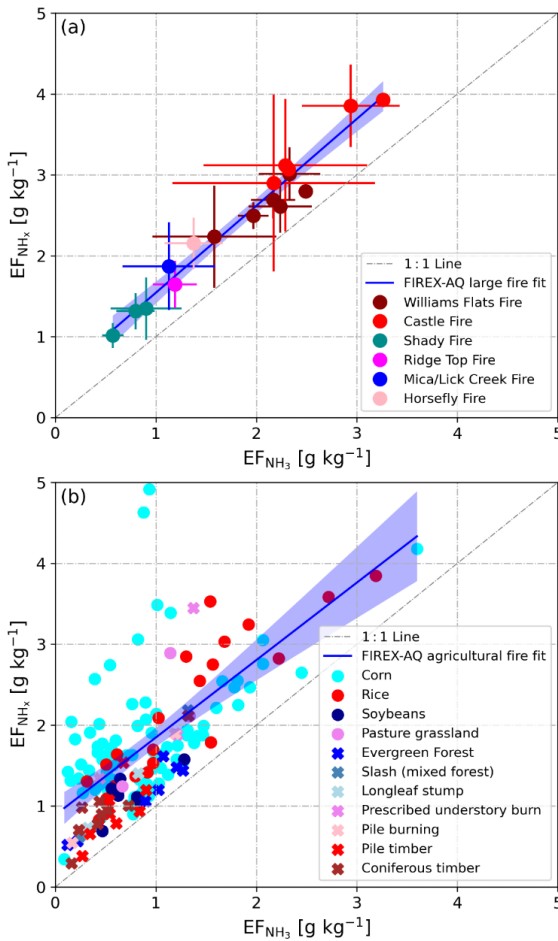

**Figure 2**: $EF_{NH_x}$ *vs.* $EF_{NH_3}$ **as derived from** *in situ* **measurements in the plumes of (a) six wildfires and (b) 66 small**

**agricultural fires (circles: field-dominated fuels, crosses: timber-dominated fuels; see section 3.3 for details)**





### 3.2 NH₃ and NHₓ emissions from wildfires in the Western US

*In situ* measurements of $NH_3$ and $NH_4^+$ were made in smoke plumes emitted from six wildfires in the Western US. Table 1 provides a detailed overview of $EF_{NH_3}$ and $EF_{NH_x}$ derived from these measurements. Plumes from the Shady, Williams Flats and Castle Fires were sampled multiple times and we list the data from each of the sampling patterns as well as the average value. $EF_{NH_3}$ and $EF_{NH_x}$ were lowest for the Shady Fire. The low emissions may be caused by the difference in fuels, which in the case of the Shady Fire was modified or managed xeric understory (Tab. S1).

Table 1: $EF_{NH_3}$ and $EF_{NH_x}$ derived from *in situ* measurements in the plumes of 6 wildfires in the Western US.

| | | | $EF_{NH_3}$ (g kg⁻¹) | | $EF_{NH_x}$ (g kg⁻¹) | | MCE | |
|---|---|---|---|---|---|---|---|---|
| Name | State | Date (dy.mo.yr) | mean | SD | mean | SD | mean | SD |
| Shady 1st pattern | ID | 25.07.2019 | 0.57 | 0.11 | 1.02 | 0.15 | 0.906 | 0.003 |
| Shady 2nd pattern | ID | 25.07.2019 | 0.80 | 0.19 | 1.32 | 0.23 | 0.892 | 0.007 |
| Shady 3rd pattern | ID | 25.07.2019 | 0.90 | 0.35 | 1.35 | 0.39 | 0.887 | 0.029 |
| **Shady mean** | | | **0.76** | **0.22** | **1.23** | **0.26** | **0.895** | **0.013** |
| Williams Flats 1st pattern | WA | 03.08.2019 | 1.58 | 0.62 | 2.24 | 0.63 | 0.908 | 0.003 |
| Williams Flats 2nd pattern | WA | 03.08.2019 | 2.24 | 0.32 | 2.61 | 0.33 | 0.907 | 0.005 |
| Williams Flats | WA | 06.08.2019 | 2.33 | 0.31 | 3.01 | 0.33 | 0.894 | 0.004 |
| Williams Flats 1st pattern | WA | 07.08.2019 | 2.49 | 0.07 | 2.80 | 0.09 | 0.905 | 0.004 |
| Williams Flats 2nd pattern | WA | 07.08.2019 | 1.97 | 0.15 | 2.50 | 0.16 | 0.909 | 0.001 |
| Williams Flats 3rd pattern | WA | 07.08.2019 | 2.17 | -- | 2.69 | -- | 0.901 | 0.001 |
| **Williams Flat mean** | | | **2.13** | **0.28** | **2.64** | **0.29** | **0.904** | **0.003** |
| Castle 1st pattern | AZ | 12.08.2019 | 2.29 | 0.81 | 3.12 | 0.82 | 0.884 | 0.003 |
| Castle 2nd pattern | AZ | 12.08.2019 | 2.94 | 0.49 | 3.85 | 0.51 | 0.890 | 0.002 |
| Castle longitudinal transect | AZ | 12.08.2019 | 2.32 | -- | 3.07 | -- | 0.864 | -- |
| Castle 1st pattern | AZ | 13.08.2019 | 2.17 | 1.01 | 2.90 | 1.09 | 0.895 | 0.007 |
| Castle 2nd pattern | AZ | 13.08.2019 | 3.26 | -- | 3.93 | -- | 0.892 | -- |
| **Castle mean** | | | **2.60** | **0.77** | **3.37** | **0.81** | **0.885** | **0.004** |
| Ridge Top | MT | 02.08.2019 | 1.19 | 0.22 | 1.65 | 0.29 | 0.940 | 0.011 |
| Mica/Lick Creek | ID | 02.08.2019 | 1.13 | 0.46 | 1.87 | 0.54 | 0.913 | 0.021 |
| Horsefly | MT | 06.08.2019 | 1.38 | 0.29 | 2.15 | 0.32 | 0.859 | 0.010 |

Average $EF_{NH_3}$ and $EF_{NH_x}$ values for the six wildfires in the Western US were 1.86 ± 0.75 g kg⁻¹ and 2.47 ± 0.80 g kg⁻¹, respectively. We compare our results to those obtained in two recent studies. Lindaas et al. (2021) investigated $NH_3$ emissions from wildfires in the Western US during the 2018 WE-CAN campaign. We calculated an average $EF_{NH_3}$ of 1.48 ± 0.91 g kg⁻





[1] for the WE-CAN data. This is slightly lower than the average $EF_{NH_3}$ reported herein, but within the combined uncertainties of the two methods: ±12% for the quantum-cascade tunable infrared laser direct absorption spectrometer (QC-TILDAS) used during WE-CAN, and ±15% for the PTR-ToF-MS analyzer used during FIREX-AQ. Selimovic et al. (2018) investigated

emissions from fires fueled by a wide range of US vegetation types in the FIREX FireLab 2016 laboratory study. We only used the data for FIREX-AQ relevant fuels (see Table S1) and obtained a significantly lower average $EF_{NH_3}$ of 0.67 ± 0.38 g kg$^{-1}$ for the FIREX FireLab data. This finding seems to confirm that laboratory fires do not realistically simulate wildfires (*e.g.*, Yokelson et al., 2013, Hodshire et al., 2019) and thereby underestimate real-world emissions of NH$_3$.

In Figure 3, we plot the measured $EF_{NH_3}$ values (6 wildfires, multiple sampling of 3 fires) as a function of MCE along with

trends from the WE-CAN and FIREX FireLab studies. In the case of the FIREX-AQ data (regression line and confidence band in blue), $EF_{NH_3}$ and MCE correlated poorly, with Pearson's coefficient of determination ($R^2$) being only 0.04. As opposed to the WE-CAN study (regression line and confidence band in black) and the FIREX FireLab experiments (regression line and confidence band in dark yellow), we did not find a clear inversion correlation between MCE and NH$_3$ emissions.

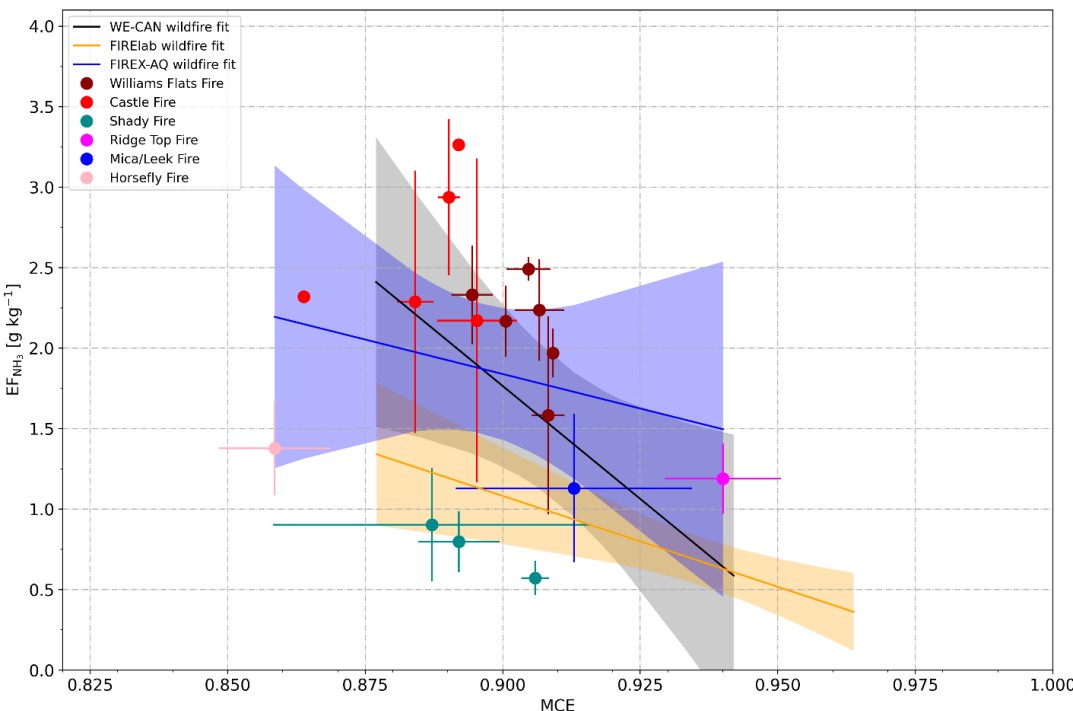

**Figure 3: Scatter plot of $EF_{NH_3}$ (as measured for six Western US wildfires during FIREX-AQ and obtained from two literature sources) *vs.* MCE.**

Finally, we would like to discuss the results presented by Gkatzelis et al. (2022), who provide an overview of EFs derived from the 2019 FIREX-AQ data. Their analysis yielded an average $EF_{NH_3}$ of 1.15 ± 0.79 g kg$^{-1}$, which is significantly lower





than the average value we have presented herein. This discrepancy can be explained by the fact that Gkatzelis et al. (2022) did

not include the additional contribution from the $NH_3$ signal tailing outside the plume, nor did it include data from multiple

plume transects downwind of the fire. The fact that different analyses of the same data result in significantly different $EF_{NH_3}$

highlights the fact that data comparisons between different studies need to be interpreted with caution.

### 3.3 $NH_3$ and $NH_x$ emissions from agricultural fires in the Southeastern US

*In situ* measurements of $NH_3$ and $NH_4^+$ were made in smoke plumes emitted from 66 small agricultural fires in the Southeastern

US. $EF_{NH_3}$ values varied widely, covering a range from 0.09 to 3.60 g kg$^{-1}$. The following average values and standard

deviations were derived: $EF_{NH_3} = 0.89 \pm 0.58$ g kg$^{-1}$, $EF_{NH_x} = 1.74 \pm 0.92$ g kg$^{-1}$, MCE = $0.92 \pm 0.04$.

We grouped the agricultural fuels into field-dominated and timber-dominated fuels. The field-dominated fuels include corn,

rice, soybeans, and grassland pasture. The timber-dominated fuels include evergreen forest, coniferous timber, prescribed

understory fire, pile burning, slash burning, pile timber slash mixture burns, and pile burning of longleaf pine tree stumps.

Table 2 lists $EF_{NH_3}$, $EF_{NH_x}$ and MCE for the two main categories and 11 subcategories.

**Table 2: $EF_{NH_3}$, $EF_{NH_x}$ and MCE as measured during FIREX-AQ for small agricultural fires in the Southeastern US burning on different types of fuel.**

| | | $EF_{NH_3}$ (g kg$^{-1}$) | | $EF_{NH_x}$ (g kg$^{-1}$) | | MCE | |
|---|---|---|---|---|---|---|---|
| **Fuel type** | 66 | mean | SD | mean | SD | mean | SD |
| Corn | 33 | 0.90 | 0.58 | 1.85 | 0.78 | 0.937 | 0.023 |
| Rice | 11 | 1.21 | 0.74 | 2.13 | 0.93 | 0.897 | 0.057 |
| Soybean | 4 | 0.75 | 0.27 | 1.16 | 0.27 | 0.925 | 0.024 |
| Grassland Pasture | 2 | 1.05 | 0.34 | 2.08 | 0.82 | 0.824 | 0.109 |
| **Field-dominated average** | **50** | **0.95** | **0.60** | **1.92** | **0.92** | **0.926** | **0.042** |
| Evergreen Forest | 1 | 0.83 | 0.47 | 1.13 | 0.44 | 0.914 | 0.027 |
| Pile burning (mixed) | 4 | 0.51 | 0.47 | 1.12 | 0.62 | 0.920 | 0.042 |
| Slash burn (mixed forest) | 2 | 0.78 | 0.76 | 1.40 | 1.10 | 0.918 | 0.063 |
| Prescribed understory burn | 1 | 1.39 | 0.19 | 2.67 | 1.11 | 0.864 | 0.022 |
| Pile timber slash | 5 | 0.58 | 0.21 | 0.95 | 0.33 | 0.858 | 0.060 |
| Pile longleaf pine tree stump | 1 | 0.57 | 0.35 | 1.07 | 0.48 | 0.877 | 0.004 |
| Timber slash coniferous | 2 | 0.51 | 0.35 | 0.98 | 0.49 | 0.912 | 0.019 |
| **Timber-dominated average** | **16** | **0.67** | **0.42** | **1.19** | **0.68** | **0.896** | **0.045** |

The data listed in Table 2 indicate that field-dominated fuels emit more $NH_3$ and $NH_x$ than timber-dominated fuels. Agricultural

areas are usually nitrogen-fertilized, which may cause increased $NH_3$ emissions. $EF_{NH_x}$ is roughly a factor of two higher than

$EF_{NH_3}$, which indicates higher primary $NH_4^+$ emissions and/or a very rapid $NH_3$ to $NH_4^+$ conversion in these fresh plumes.



Figure 4 shows $EF_{NH_3}$ as a function of MCE for different fuels (as measured in individual fires), the averages derived for field-dominated fuels and timber-dominated fuels and the results from four previous studies (McMeeking et al., 2009; Stockwell et al., 2015; Müller et al., 2016; Selimovic et al., 2018). Also in this case, $EF_{NH_3}$ and MCE correlated poorly with $R^2$ being 0.05.

The literature values match the low $NH_3$ emissions (< 1 g kg$^{-1}$) we observed for most agricultural fires, but the high $NH_3$ emissions from burning rice and corn residues have not been reported before.

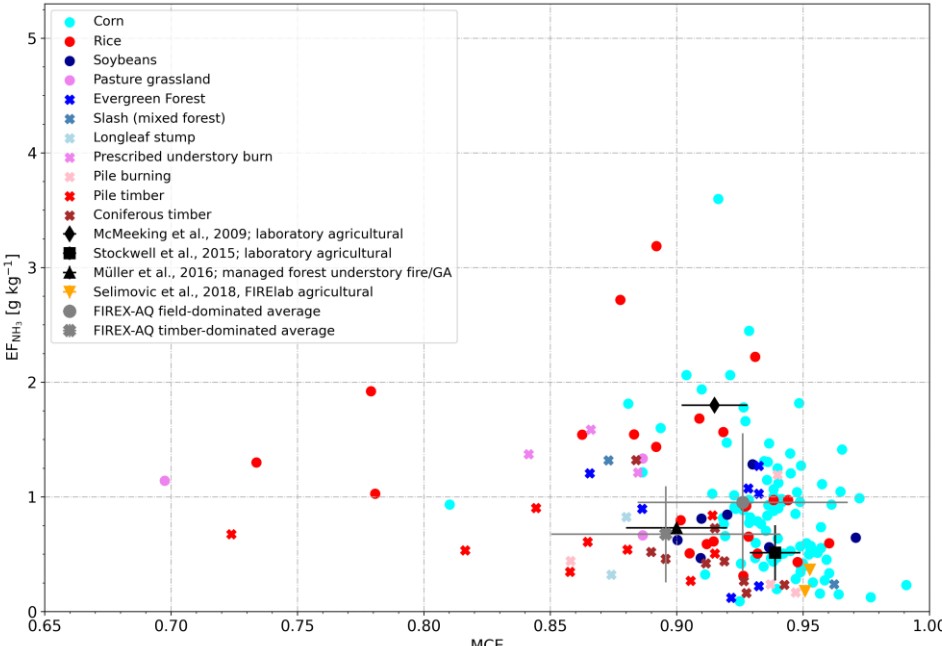

**Figure 4: Scatter plot of $EF_{NH_3}$ *vs*. MCE as measured for 66 agricultural fires (circles: field-dominated fuels, crosses: timber-dominated fuels) and obtained from four literature sources.**

**3.4 NH₃ and NHₓ emissions from other fires**

We also measured $NH_3$ and $NH_4^+$ in smoke plumes emanating from two prescribed grassland fires in the Central US and one prescribed forest fire in the Southern US. These fires do not fall within the two main categories discussed in the previous two sections and are thus separately presented here. Table 3 lists the fire details, $EF_{NH_3}$, $EF_{NH_x}$ and MCE for these three fires.



**Table 3: $EF_{NH_3}$, $EF_{NH_x}$ and MCE as measured during FIREX-AQ for two prescribed grassland fires in the Central US and one prescribed forest fire in the Southern US.**

| Fire details | $EF_{NH_3}$ (g kg$^{-1}$) | | $EF_{NH_x}$ (g kg$^{-1}$) | | MCE | |
|---|---|---|---|---|---|---|
| | mean | SD | mean | SD | mean | SD |
| Hickory Ridge State Wildlife Management Area prescribed, 29.08.2019, NE grass during the green growing season (not dry) | 0.33 | 0.12 | 1.19 | 0.93 | 0.907 | 0.002 |
| Tallgrass Prairie National Preserve prescribed, 29.08.2019, KS prairie tallgrass during the green growing season (not dry) | 0.92 | 0.20 | 1.73 | 0.49 | 0.894 | 0.014 |
| Black Water River State Forest prescribed, 30.08.2019, FL oak, mature longleaf pine, mesic and xeric shrub, grass, litter, understory | 0.30 | 0.16 | 0.61 | 0.20 | 0.942 | 0.006 |

## 4 Conclusions

During the 2019 FIREX-AQ field campaign, we measured NH$_3$ and NH$_4^+$ aboard the NASA DC-8 in wildfire and agricultural fire plumes. We found that NH$_4^+$ was either directly emitted from the fire (consistent with past laboratory experiments) and/or NH$_3$ had already partially partitioned to particulate NH$_4^+$ at the time of sampling. We thus also evaluated emissions of NH$_x$ and produced a comprehensive set of $EF_{NH_3}$ and $EF_{NH_x}$ for wildfires in the Western US and agricultural fires in the Southeastern US. Our data show no clear inverse correlation between MCE and $EF_{NH_3}$. $EF_{NH_3}$ values measured in plumes of large wildfires were similar to measurements of past field studies, but significantly higher than observed in prior work on laboratory simulated fires, which typically burn on a single fuel. We also report the first extensive set of field measurement derived $EF_{NH_3}$ and $EF_{NH_x}$ values for different types of agricultural fires in the Southeastern US. NH$_3$ emissions were highest from fires of corn and rice residues, which may be caused by fertilization of these fields. Substantial amounts of NH$_4^+$ were detected in freshly emitted smoke from some of the corn and rice field fires, which warrants further investigation.

## Data availability

All the FIREX-AQ data are available at NASA's Atmospheric Science Data Center (DOI: 10.5067/SUBORBITAL/FIREXAQ2019/DATA001) (NASA, 2019).

## Author contribution

LT supported the PTR-ToF-MS instrument development, performed the field measurements, performed the data analysis and interpretation, and wrote the manuscript draft. TM built and characterized the modified PTR-ToF-MS instrument and



performed field measurements. FP performed field and laboratory measurements and supported the data analysis as well as the PTR-ToF-MS instrument development. CJN supported the post-mission calibration experiments. PCJ, HG, BAN, MSK and JLJ provided the $NH_4^+$ data. HH, GD, JPD, JBN provided the $CO_2$, CO and $CH_4$ data. EBW, EG and AJS provided the fuel characterization information. AW conceived the modified PTR-ToF-MS instrument, supervised the measurements and data analysis, performed field measurements and finalized the manuscript. All authors commented and accepted the final
version of the manuscript.

**Conflicts of interest**

The authors declare no conflict of interest.

**Acknowledgements**

Laura Tomsche's research was supported by an appointment to the NASA Postdoctoral Program at the NASA Langley Research Center, administered by Universities Space Research Association under contract with NASA. The FIREX AQ project was funded by the NASA Tropospheric Composition Program (TCP). The University of Innsbruck PTR-ToF-MS instrument was partially funded by the Austrian Federal Ministry for Transport, Innovation and Technology (bmvit, FFG, ASAP). FP received funding from the European Union's Horizon 2020 research and innovation program under grant agreement no. 674911
(IMPACT). PCJ, HG, BAN, MKS, and JLJ were supported by NASA Grants 80NSSC18K0630 and 80NSSC21K1451.

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
