# Peer review of "Measurement report: Emission factors of $NH_3$ and $NH_x$ for wildfires and agricultural fires in the United States"

_EGUsphere, 2022_

## Author Comment (AC1)

We thank reviewer #1 for having carefully read our manuscript and for making very valuable comments and suggestions. Here is how these were addressed:

***The only major issue is stated in line 100 - in field calibrations were inconsistent. It's not clear how much of an issue this is, but anytime a calibration fails in situ one is concerned about the quality of the data. The authors need to provide more detail so this concern can be alleviated. How confident are they that the measurements are accurate based on the laboratory calibration.***

This should indeed have been explained better. We now provide more details and the modified text reads as follows:

> "We performed three types of calibrations in the field: i) periodic in-flight calibrations using a dynamically diluted calibration standard in a pressurized cylinder (10 ppm $NH_3$ in $N_2$; Praxair Distribution Inc., Lancaster, CA, U.S.A.), ii) a ground-based calibration using a dynamically diluted calibration standard in a pressurized cylinder (2.7 ppm $NH_3$ in $N_2$; provided by NOAA's Chemical Sciences Laboratory), iii) a ground-based calibration using an $NH_3$ permeation source (provided by NOAA's Chemical Sciences Laboratory). While results from the cylinder-based calibrations were in good agreement, the permeation tube based calibration yielded an a factor of two higher instrumental response factor. For resolving this inconsistency, we carried out an extensive post-mission $NH_3$ calibration in the laboratory. For that purpose, an artificial atmosphere ($NH_3$ in air) was generated in a 250 L environmental ("smog") chamber equipped with a Fourier Transform Infrared (FT-IR) spectrometer (Bruker IFS 66v/S). The concentrations of $NH_3$ (accuracy: ±5 %) were determined from the FT-IR spectra (120 m path length, 0.125 $cm^{-1}$ spectral resolution) in a global non-linear least squares spectral fitting procedure (Griffith, 1996) employing the absolute cross sections of $NH_3$ (Gordon et al., 2017). The instrumental response factor derived from the post-mission calibration agreed to within ±15 % with the response factors obtained during the in-flight calibrations. We concluded that the permeation rate determined in pre-campaign laboratory measurements had changed in the field. The estimated accuracy of the reported $NH_3$ mixing ratios is ±15 %. We note that this accuracy estimate is not valid when $NH_3$ mixing ratios abruptly changed and inlet/instrument surfaces were not equilibrated."

***In the introduction, line 59- it would be useful to provide more information to the reader on the differences of the Innsbruck instrument to the other instruments discussed in the paragraph.***

No other instruments are actually discussed in this paragraph. For making this clear, we now refer to the "conventional PTR-ToF-MS instrument" used in the cited previous studies and the "modified PTR-ToF-MS instrument "used during FIREX-AQ. The modifications are described in section 2.2.

***Section 2.2 - The authors should provide some evidence to support the statement on line 87 that the instrumental background was reduced to single digit ppb levels.***

This is actually not easy to show in a simple plot. The instrumental background consisted of a "chemical" background (*i.e.*, $NH_3$ formed in the plasma source) and a "mass spectral" background (*i.e.*, a peak feature in that region of the mass spectrum that caused an enhanced baseline at *m/z* 18.034). The latter was peculiar to the instrument/operational settings used during FIREX-AQ. We did not deconvolute the two contributions in our automated data analysis. A manual reanalysis of selected mass spectra shows that the chemical background ranged from sub-ppb levels (observed under optimized conditions in the laboratory) to single and sometimes even double-digit-ppb levels (observed in the field). We feel that a measurement report does not need to go into such fine instrumental details. We have thus removed the sentence in

which we make a quantitative statement. In our opinion it is sufficient to qualitatively describe that He addition to the source drift region effectively suppresses $NH_3$ formation in the plasma. It will anyway vary from instrument to instrument how low the chemical background will get.

**Line 111 - what are the implications of the 34% uncertainty in the NH4+ data on the final calculated emissions factors? Can that uncertainty somehow be propagated through?**

Fire emission factors have a high natural variability and we feel it is important to show the extent of this variability, without confounding it with the measurement uncertainty (and the uncertainty in the carbon fraction). We do, however, explicitly mention the uncertainties of the underlying measurement variables ($NH_3$, $NH_4^+$, $CO_2$, $CO$, $CH_4$), providing the reader with all measurement uncertainty information. The measurement uncertainty in $NH_4^+$ actually has only a small impact on the uncertainty of $NH_x$: an average $NH_4^+$ fraction of 29% converts into a 15% uncertainty for $NH_x$, the maximum $NH_4^+$ fraction of 52% converts into a 19% uncertainty for $NH_x$.

**Line 114 - There is potentially a lot of organic N compounds in burning emissions (e.g., Mace et al., 2003 Water-soluble organic nitrogen in Amazon Basin aerosols during the dry (biomass burning) and wet seasons). This could be contributing to the large amount of NH4+ measured early on in the fires. This needs to be stated very clearly throughout the paper as the authors have no way of knowing if this is an issue or not.**

This is clearly stated in section 2.2 of the original manuscript: "We note that, based on the current state of knowledge, the AMS $NH_4^+$ data collected in fresh smoke plumes suffer from a minor (≤ 20%) positive interference from reduced organic nitrogen compounds. A general correction is still under development based on positive matrix factorization (PMF) analysis." In the revised manuscript, we have added a subsentence and the given reference: "…reduced organic nitrogen compounds, which are known to be abundant constituents of biomass burning particles (*e.g.*, Mace et al., 2003)." We also reiterate our comment in the Results section (3.1): "As stated in section 2.2, the $NH_4^+$ measurement suffer from a minor (≤ 20%) positive interference from reduced organic nitrogen compounds."

**Section 2.3 - is this the most commonly used method for calculating EFNH3? Also, why was the carbon fraction assumed to be 0.5? This should be justified.**

Yes, this is how emission factors are usually calculated, although we are using a simplified formula that only applies to $NH_3$.

We have added a brief explanation on the carbon fraction:
"$F_c$ is the fraction of carbon in the fuel, which is typically in the 0.45-to-0.55 range (Akagi e al., 2011 and references therein). We assumed $F_c$ to be 0.50 and note here that the resulting 10% uncertainty in $EF_{NH_3}$ is small compared to the natural variability of $EF_{NH_3}$."

**Conclusions on line 215 - If the discrepancy is based on the tailing and downwind data, then the authors could prove that by doing the calculation on their data excluding tailing and the downwind data. This would provide confidence in the conclusions.**

The Gkatzelis et al. manuscript is still in preparation. We have deleted this paragraph and reference, as suggested by the other reviewer.

---

## Author Comment (AC2)

We also thank reviewer #2 for having carefully read our manuscript and for making very valuable comments and suggestions. Here is how these were addressed:

***The EFs are reported as X ± Y g·kg⁻¹. It is unclear what the plus/minus value refers to. Is this 1σ or some other value? Does this reflect the variability across multiple transects alone, the uncertainties in variables and assumed values in the EF calculation, or both? This needs to be clearly described in the Methods. [Also, Reviewer #1's comment on propagated uncertainties.]***

This is already mentioned in the original version of the manuscript: "We thus included all plume transects in our analysis, up to where $\Delta NH_3/\Delta CO$ reached its maximum and derived an average $EF_{NH_3}$ and $EF_{NH_x}$ ($\pm$ standard deviation, SD)." We have added one sentence for further clarification: "All SDs reported herein only include the measured variability and do not consider measurement uncertainties in the underlying variables ($NH_3$, $NH_4^+$, $CO_2$, $CO$, $CH_4$)."

***An assumed carbon fraction of 0.5 for biomass may be conservatively off by ±10% (please also cite literature justifying this assumption). [Also noted by Reviewer #1]***

We have added a brief explanation on the carbon fraction:
 "$F_c$ is the fraction of carbon in the fuel, which is typically in the 0.45-to-0.55 range (Akagi e al., 2011 and references therein). We assumed $F_c$ to be 0.50 and note here that the resulting 10% uncertainty in $EF_{NH_3}$ is small compared to the natural variability of $EF_{NH_3}$."

***How was the background mixing ratio estimated? What about during agricultural fires, where there may have been numerous such occurring around the same time and around the same place contributing to NH₃/NH₄⁺ in the sampled air; can elevated VMRs specific to the plume under consideration be reliably obtained?***

The $NH_3$ values measured immediately before the plume encounter were used as background mixing ratio. The modified text reads as follows:
 "For calculating $\Delta$, we thus applied the method described in the Supplement of Müller et al. (2016) and calculated cumulative volume mixing ratios including the immediate period (10 s) before the plume was encountered (background) and the period after the plume encounter (seconds to minutes) when the $NH_3$ signal tailed off."

In 2019, the fire activity was actually very low. We thus almost exclusively sampled isolated fires. In other words, the background was negligible.

**Which plume transect segments were used in the analyses? Ideally, also plot the selected transects on the map showing location of fires (at least for the wildfires).**

This is already explained in the manuscript: "We thus included all plume transects in our analysis, up to where $\Delta NH_3/\Delta CO$ reached its maximum and derived an average $EF_{NH_3}$ and $EF_{NH_x}$ ($\pm$ standard deviation, SD)." Fig. 3 (formerly Fig. S3) now includes a map, as suggested by the reviewer.

***How were the classifications by fuel types obtained?***

This was described in section 2.2 of the original manuscript: "Information about fuel types was obtained from the 30 m Fuel Characteristic Classification System (FCCS; Ottmar et al., 2007), the 30 m Cropland Data Layer classification 2019 dataset, and ground intelligence." This was obviously misplaced and we have moved it to section 2.1.

*Lines 212-217: Since Gkatzelis et al. (2022) cited here is unpublished, unavailable as a preprint, and not made available at the time of the review, these lines cannot be reviewed. Suggest deleting or making Gkatzelis et al. (2022) available during the next round of revisions.*

The Gkatzelis et al. manuscript is still in preparation. We have deleted this reference.

**Suggest avoiding speculative statements such as: (Conclusions) "NH3 emissions were highest from fires of corn and rice residues, which may be caused by fertilization of these fields." There was no analysis presented to make this claim (even if softened by "may").**

We have removed this subsentence.

**The primary finding is that EF$_{NH3}$ may be underestimated in laboratory studies. More details are required to develop confidence in the reader that this is the case. It may be that in the "real world conditions" there is burning/heating of duff and/or of the soil itself (NH$_4^+$ and NH$_4^+$ → NH$_3$) that is not considered in the lab among other possible factors. That these are insignificant aspects and that lab- and aircraft-derived EFs are equivalent but not equal needs to be demonstrated before the finding is presented as strongly as it is in the manuscript.**

In the revised manuscript, we just describe the findings and do not draw any strong conclusions:

Abstract: "NH$_3$ emissions in ambient sampling were significantly higher than observed in previous laboratory experiments in the FIREX FireLab 2016 study."

Conclusions: "$EF_{NH_3}$ values measured in plumes of large wildfires were similar to those observed during the 2018 WE-CAN field campaign, but significantly higher than observed in the FIREX FireLab 2016 laboratory study."

**Technical corrections**

*Throughout the manuscript: ppb or ppbV and ppm or ppmV?*

We think that "ppb" is good for NH$_3$. The distinction between ppbV and ppbC is important for organic compounds.

**The supplementary figures and tables are important for the narrative. Suggest moving these to the main manuscript and not relegating to the supplement.**

We have moved the figures to the main manuscript, but not the table. The detailed fuel type description is not important for the narrative.

**Figure 1: The error shading seems to consider only the average values (circles) and not their error ranges as well. Why? Please also describe what exactly the shading represents in the figure caption.**

We have updated the figure caption.

**Suggestion: Due to wide range of values in Figs. 1 and 2b, suggestion to use log-log scale**

The small values on the y-axis of Fig. 1 are actually not relevant for the calculation of the cumulative $\Delta$. We think a linear scale more clearly visualizes the dimension of the tailing problem.

Fig. 2b: Again, we think that is more important to show the full range of $EF_{NH_3}$ and $EF_{NH_x}$ on a linear scale, rather than emphasizing differences at low values on a logarithmic scale.